# Modification of Ginseng Insoluble Dietary Fiber by Enzymatic Method: Structural, Rheological, Thermal and Functional Properties

**DOI:** 10.3390/foods12142809

**Published:** 2023-07-24

**Authors:** Guihun Jiang, Karna Ramachandraiah, Chaoyi Tan, Nanjie Cai, Kashif Ameer, Xiaoyu Feng

**Affiliations:** 1School of Public Health, Jilin Medical University, Jilin 132013, China; 15947961862@163.com (C.T.); aykr4214082@163.com (N.C.); fengxiaoyu0623@163.com (X.F.); 2School of Lifesciences, Sejong University, Seoul 05006, Republic of Korea; karna@sejong.ac.kr; 3Institute of Food Science and Nutrition, University of Sargodha, Sargodha 40100, Pakistan

**Keywords:** insoluble dietary fiber, ginseng residue, enzymatic treatment, physicochemical properties, functional characteristics, hydration properties

## Abstract

In this study, the effects of enzymatic modification using cellulase/xylanase on the composition and structural and functional properties of ginseng insoluble dietary fiber (G-IDF) were evaluated. Fourier transform infrared spectroscopy and scanning electron microcopy showed that enzymatic extraction treatment caused obvious structural alterations in ginseng-modified (G-MIDF) samples, which exhibited more porous and completely wrinkled surfaces. Comparing the peak morphology of G-MIDF with untreated IDF using X-ray diffractometry, the G-MIDF sample exhibited split peaks at a 2θ angle of 23.71°, along with the emergence of sharp peaks at 28.02°, 31.78°, and 35.07°. Thermo-gravimetric analysis showed that G-MIDF exhibited a specified range of pyrolysis temperature and is suitable for food applications involving processing at temperatures below 300 °C. Overall, it was evident from rheograms that both G-IDF and G-MIDF exhibited a resemblance with respect to viscosity changes as a function of the shear rate. Enzymatic treatment led to significant (*p* < 0.05) improvement in water holding, oil retention, water swelling, nitrite ion binding, bile acid binding, cholesterol absorption, and glucose absorption capacities.

## 1. Introduction

Ginseng (*Panax ginseng* C.A. Meyer) is a perennial plant that belongs to the Araliaceae family and is native to the mountainous regions of eastern Asia, including China, Korea, and Siberia [1]. It is highly regarded for its medicinal properties and has been used for centuries in traditional Chinese medicine. Ginseng is known for its distinctive fleshy root, which is often harvested and used in herbal remedies and supplements [2]. The root is characterized by its forked shape and light yellow or beige color. Ginseng is believed to possess various health benefits, including boosting energy levels, enhancing cognitive function, reducing stress, and promoting overall well-being [3,4]. Its adaptogenic properties are thought to help the body cope with physical and mental stressors, making it a popular natural remedy in many cultures around the world [5].

Dietary fiber provides numerous health benefits in the human body, including improved digestive health by preventing constipation and promoting regular bowel movements, aiding in weight management by increasing satiety and reducing calorie intake, and helping control blood sugar levels by slowing down the absorption of sugars [3]. Additionally, fiber contributes to heart health by reducing cholesterol levels and lowering the risk of cardiovascular diseases. Its ability to promote a healthy gut microbiome also supports overall immune function and reduces the risk of certain diseases, such as colorectal cancer [6]. Dietary supplements hold a prominent position in the global ginseng market, playing a crucial role in its growth. In 2019, the ginseng extract market reached a significant value of USD 24.5 million, with projections indicating a compound annual growth rate (CAGR) of 6.3% until 2027 [7]. This highlights the significance of dietary supplements in driving the expansion of the ginseng market on a global scale.

Dietary fibers (DFs) are complex carbohydrate polymers composed of at least ten monomeric units, resistant to hydrolysis by the body’s own enzymes in the small intestine. Insoluble fiber (IDF) is part of the composition of the DFs, and IDF has been reported to comprise several functional groups including phenolics, carboxylic acids, ketones and aldehydes [8]. IDF has been found to offer various beneficial effects such as promoting intestinal peristalsis and increases in the fecal volume and providing help to eliminate heavy metals and other toxins. While soluble fibers (SDF) also provide numerous biological, physiological, and antioxidant activities, the main constituents of DFs are primarily IDF, which should be targeted for effective utilization [9]. Consequently, modifying IDF is considered a viable nutritional strategy to harness the potential benefits of dietary fibers. Ginseng insoluble dietary fiber (IDF) is a type of dietary fiber that is found in ginseng. 

Ginseng IDF is a non-starch polysaccharide that is made up of cellulose, hemicellulose, and lignin. Ginseng IDF has a number of health benefits. Ginseng IDF can help to regulate bowel movements and promote the growth of beneficial bacteria in the gut. This can help to improve digestion and reduce the risk of constipation, diarrhea, and other digestive problems [10]. Ginseng IDF can help to lower LDL (bad) cholesterol levels and raise HDL (good) cholesterol levels. This can help to reduce the risk of heart disease, stroke, and other cardiovascular problems. Ginseng IDF can help to slow down the absorption of glucose into the bloodstream. This can help to keep blood sugar levels stable and reduce the risk of type 2 diabetes [11]. Ginseng IDF can help to boost the immune system by stimulating the production of white blood cells. This can help to protect the body against infection. Ginseng IDF can help to reduce inflammation by blocking the production of pro-inflammatory cytokines. This can help to improve symptoms of conditions such as arthritis, asthma, and allergies [12]. In addition to these health benefits, ginseng IDF is also a good source of fiber, which is essential for overall health. Fiber helps to keep the digestive system healthy, promote weight loss, and regulate blood sugar levels. Ginseng IDF is a safe and effective dietary supplement that can be added to the diet in a number of ways. It can be found in capsules, powders, and teas. It can also be added to smoothies, yogurt, and other foods [13]. Ginseng IDF may interact with certain medications, so it is important to make sure that it is safe for you to take. Overall, ginseng IDF is a beneficial dietary fiber that has a number of health benefits. It can help to improve gut health, reduce cholesterol levels, lower blood sugar levels, boost the immune system, and reduce inflammation. If you are looking for a way to improve your health, adding ginseng IDF to your diet is a good option [14].

The inclusion of insoluble dietary fibers (IDF) in food products can lead to quality problems in terms of flavor, color, and texture. To address these issues, various methods for modifying fibers like enzymatic, fermentation, as well as chemical processes have been reported in the published literature. Chemical extraction methods, including acidic, alkaline, chelating, and oxidant extractions are commonly utilized to obtain dietary fibers (DFs), with the method selection being dependent on the desired polysaccharide [15]. Additionally, research has demonstrated that the quantity and composition of ginsenosides, a type of saponins, vary depending on the extraction and analytical methods employed. However, it should be noted that strong alkaline conditions may cause disruption in glycosidic linkages in DFs, thereby exerting an influence on the balance between IDFs and soluble dietary fibers (SDFs). Therefore, it is the need of the time to extract ginseng DF with beneficial health effects, which necessitates carrying out detailed studies of the main ginseng component termed as ginseng IDF [16]. To the best of our knowledge, no reports are available so far regarding the application of enzymatic modification methods of ginseng IDF. Therefore, this study aimed to assess the effects of enzymatic modification using cellulase/xylanase on the composition and structural and functional properties of ginseng IDF.

## 2. Materials and Methods

### 2.1. Sample Procurement

From a local supermarket in Jilin, China, dried ginseng roots were purchased. The ginseng roots were subject to cultivation for a period of 5 years, and procurement was carried out in the season of November 2022. 

### 2.2. Preparation of Ginseng Residue

The preparation of ginseng residue was carried out following the method of Jiang et al. [1]. The polysaccharides extraction from the ginseng residue was performed by boiling, and the obtained residue was subjected to washing with ethanol and subsequently with distilled water in order to eliminate the oligosaccharides (water-soluble) and inorganic salts. The ginseng residue was subjected to drying at 60 °C for 24 h and sieving (60 mesh). The dried residue was subjected to packaging using a self-sealing bag and was then stored at a temperature of −20 °C until further analysis.

### 2.3. Extraction of Ginseng IDF 

IDF preparation was performed using ginseng residue, as described by Kurek et al. [17], with slight modification. The treatment of ginseng residue was carried out by means of heat-stable α-amylase and protease in order to achieve the removal of proteins and of starches. Then, the drying of precipitate was carried out at a drying temperature of 50 °C for period of 3 h to bring about the derivation of IDF. The dried residue was subjected to packaging using a self-sealing bag and then was stored at the temperature of −20 °C until further analysis. 

### 2.4. Preparation and Modification of Ginseng IDF 

The modification of ginseng IDF was carried out based on a modified method described by Ma et al. [18]. Ginseng IDF was subjected to rigorous mixing using a phosphate buffer solution (PBS, *w*/*v* = 1:20) for pH value adjustment to 4.9. Then, 0.25% cellulase/xylanase solution (*w*/*v*, %; cellulase (U): xylanase (U) = 1:1) was taken and subjected to rigorous mixing with PBS at a temperature of 60 °C in an oscillating water bath for hydrolyzation for 1.5 h. For the inactivation of the enzymes, the mixture solution was subjected to boiling for a period of 15 min. The solution was subjected to centrifugation for a time period of 10 min at the centrifugation rate of 4000× *g*. The, the enzymatically modified IDF (MIDF) was utilized to extract the residue and then subjected to washing and drying. 

### 2.5. Scanning Electron Microscopy (SEM) 

The morphological attributes and microstructural properties of IDF and its MIDF were evaluated by means of SEM (SU8010, Hitachi, Tokyo, Japan). The samples were subjected to slicing into rectangular shapes by keeping them on the horizontal plane. The sample was mounted on a specimen holder, followed by gold sputter-coating. The microstructural attributes were analyzed using SEM (at an operational 5 kV voltage). Capturing of the SEM imagery was performed at magnification levels of 1000× and 2000×.

### 2.6. Fourier Transform Infrared (FT-IR) Spectroscopy 

An FT-IR spectrophotometer (FTIR E4, FTIR Systems Ltd., Kent, UK) was employed for analyzing the organic functional groups present in ginseng IDF as well as MIDF samples. The FT-IR spectra were recorded at the wavelength range of 400–4000 cm^−1^ at 4 cm^−1^ resolution at <10 s scan speed. The FT-IR attenuated total reflection (ATR) mode was employed to obtain FT-IR spectra. 

### 2.7. X-ray Diffraction (XRD) 

The XRD profiles of IDF and MIDF samples were determined by means of a diffractometer (PANalytical X’Pert PRO, Almelo, The Netherlands). The equipment specifications for analysis were as follows: diffraction angle (2θ) falling in the range of 4–40°, as well as copper Kα radiation (0.154 nm, 40 kV, 50 mA).

### 2.8. Thermal Properties 

A thermogravimetric analyzer (TGA 550, TA Instruments, Newcastle, DE, USA) was utilized for the evaluation of ginseng IDF samples (5 mg). The temperature was in the range of 30–600 °C, whereas the heating rate was set at a rate of 20 °C/min under nitrogen atmosphere.

### 2.9. Rheological Properties 

#### 2.9.1. Solution Preparation 

The dissolution of IDF samples in sodium phosphate buffer (20 mM, pH 6.5) was carried out for the preparation of 1% solutions mixture in terms of *w*/*v* on basis of dry weight by employing a magnetic stirrer at ambient temperature for 10 min. The heating temperature of the solution was elevated to 80 °C, aided by constant stirring and temperature maintenance for a time period of 15 min. After reaching the heating temperature, the solution mixture was subjected to cooling with constant stirring to an ambient temperature. Finally, the solutions were stored at a temperature of 4 °C for 24 h.

#### 2.9.2. Rheological Measurements 

Rheological measurements were carried out for SDF solutions by employing a rheometer (Discovery HR-1, TA, Newcastle, DE, USA). Steel cone geometry with a diameter of 60 mm and gap of 56 µm was utilized to perform the analysis. Continuous shear test was applied with a shear rate ranging from 0.10 to 1000 s^−1^ at an operational temperature of 25 °C. With a corresponding rising trend of shear rate ranging from 0.10 to 1000 s^−1^, the measurement of sample viscosities was performed. All samples were assessed thrice. IDF and MIDF were evaluated in terms of apparent viscosity (η), consistency coefficient (K), and flow behavior index (n). The experimental data were subjected to fitting in accordance with the power law model equation given below:η = Kγ n^−1^(1)
where apparent viscosity (Pa·s^−^^1^) was denoted by η, consistency coefficient (Pa·s n) by K and the flow behavior index by n.

### 2.10. Hydration Properties 

The water holding, (WHC), oil holding (OHC), and water swelling capacities (WSC) were evaluated in accordance with the method reported by Meng et al. [19]. In brief, a specified amount of a sample, weighed to 0.5 g, was placed in a 10 mL centrifuge tube. An amount of 5 mL of distilled water was mixed with the samples, followed by incubation for a 1 h time interval at 37 °C. Then, the water was separated through filtration, and then WHC of IDF and MIDF samples were assessed using the equation given below:WHC (g/g) = (W_2_ − W_1_)/W_1_(2)
where W_1_ denotes to sample total dry weight and W_2_ is the sample total weight after water has been filtered out. 

OHC assessment of IDF and MIDF was performed. In brief, a sample accurately weighed to 0.5 g was subjected to mixing with soybean oil in a centrifuge tube followed by incubation for 1 h at a temperature of 37 °C. Then, the sample was subjected to a process of centrifugation at 5000× *g* for a time interval of 10 min. Following centrifugation the centrifuge tube was inverted in order to discard excessive amount of soybean oil. The OHC was then quantified as per the equation given below: OHC (g/g) = (W_2_ − W_1_)/W_1_(3)
where the sample’s total dry weight is denoted by W_1_ and the sample’s total weight calculated after discarding excess oil is denoted by W_2_. A sample accurately weighed to a specified amount of 0.5 g was added to the centrifuge tube (10 mL) and subjected to mixing with distilled water (5 mL). Then, the reaction mixture was added for incubation for 24 h at a temperature of 37 °C. Then, WSC was assessed as per the equation given below: WSC = (V_2_ − V_1_)/W(4)
where sample volume prior to hydration (mL) is denoted by V_1_, V_2_ is indicative of hydrated sample volume (mL), and DF weight prior to hydration (g) is represented by W. 

### 2.11. Nitrite Adsorption Capacity (NAC) 

The method reported by Luo et al. [20] was employed for the quantification of the NAC. In brief, an IDF sample weighed to 1 g was subjected to mixing with sodium nitrite solution (100 μmol/L), accurately measured to total volume of 25 mL. The pH of the sample was adjusted to a final pH of 2.0 (resembling pH conditions of the stomach) or pH 7.0 (simulating the pH conditions of the small intestine). Then, incubation of the sample was carried out at a temperature of 37 °C for a time interval of 120 min. Determination of the nitrite content was performed using N-(1-Naphthyl) ethylenediamine dihydrochloride/sulfanilic acid reagent at a spectrophotometric wavelength of 538 nm (UV-1800, Shimadzu Instr. Mfg. Co., Ltd., Kyoto, Japan). 

### 2.12. Bile Acid Adsorption Capacity (BAC) 

Measurement of the BAC was performed in accordance with the method of Luo et al. [20]. In brief, an IDF sample weighed to 1 g was subjected to mixing with 30 µL of sodium cholate (1–3 mg/mL, pH 7.0) and incubation for a time interval of 2 h at a temperature of 37 °C. Supernatant absorbance was assessed by means of a spectrophotometer at 620 nm wavelength. Then, the BAC was quantified and expressed in terms of mg/g.

### 2.13. Cholesterol Adsorption Capacity (CAC) 

For calculation of the CAC, the reported method by Luo et al. [20] was employed. In brief, an IDF sample weighed to 1 g was subjected to mixing with a diluted yolk solution with a 30 mL volume (using 9 mL volume of distilled water, a fresh egg yolk was diluted, followed by homogenization) [21]. After pH adjustment to 7.0, samples were incubated for a time interval of 180 min at a temperature of 37 °C. Measurement of the absorbance value of the supernatant was assessed at a wavelength of 550 nm. Then, the CAC was quantified and expressed in terms of mg/g. 

### 2.14. Glucose Adsorption Capacity (GAC) 

In brief, an IDF sample accurately weighed to 0.5 g was subjected to mixing with 50 mL glucose solutions (at concentrations of 50, 100, and 150 mmol/L). After being subjected to rigorous stirring in a continuous manner for a time interval of 2 h at a temperature of 37 °C, samples were centrifuged for a period of 5 min at a centrifugation speed of 2862× *g*. Determination of the final glucose concentration of the collected supernatant was performed by means of a glucose (oxidaseperoxidase) assay kit (Rongsheng Biotech Co., Ltd., Shanghai, China). Then, the GAC was quantified and expressed in terms of mg/g. 

### 2.15. Statistical Analysis 

All experiments were performed in triplicate. One-way analysis of variance (ANOVA) was employed to assess the differences between the treatments using SPSS version 18.0 (Chicago, IL, USA). Mean differences were determined by Tukey’s test at a confidence level of 95% (*p* < 0.05). 

## 3. Results and Discussion

### 3.1. Structural Analysis of Enzymatically Modified Ginseng Residue Dietary Fiber

#### 3.1.1. SEM Microstructure 

SEM images of ginseng insoluble dietary fiber (G-IDF) and enzymatically modified ginseng insoluble dietary fiber (G-MIDF) are demonstrated in Figure 1 at magnification levels of 1000× and 2000×. Micrographs of G-IDF (Figure 1A–C) showed microstructural attributes. A loosely packed microstructure was more evident on the surface of the G-IDF. A smooth surface was more visible in the case of the G-IDF micrographs taken at 2000× magnification. Cracks were also present on the smooth surface of the G-IDF. Moreover, the G-IDF appeared to be more compact, denser, and exhibited irregular shapes. Contrary to this, enzymatic extraction treatment caused significant structural alterations in the G-MIDF. The G-MIDF exhibited a more porous and completely wrinkled surface. This may be attributed to the looser inner structural framework of the G-MIDF, which was evidence of the fact that enzymatic treatment caused breakage in the glycosidic bonds in the IDF [22]. The hydrolysis caused by enzymatic actions led to damage to the microstructure, resulting in the formation of holes in the IDF’s structural framework (Figure 1B–D). Cellulose and hemicellulose are regarded as the major cell wall components. Lignin is usually found throughout the plant matrix and is highly concentrated in middle lamella. Plant tissues are characterized in terms of cellulose embedded in a matrix composed of proteins, pectins, and hemicelluloses [3]. Hemicelluloses are usually found in bound form with both lignin and microfibrils via hydrogen bonds. Enzymatic treatment caused the degradation of lignin, which resulted in a weakening of cellulosic and hemicellulosic structures and led to their partial degradation. In a similar manner, decreasing tendencies in lignin and hemicelluloses have been reported in previously published reports [23,24]. While lignin and hemicellulose exhibited decreases, cellulose content showed increases, and this may be attributed to the small molecules of other polysaccharides or non-fiber contents. The appearance of cracks and fissures was evident on the surface of the G-MIDF, and this could be due to the hydrolysis effect of the enzymatic reaction resulting in the loss of water-soluble substances from the ginseng residue DF [25]. This also resulted in structural destruction in the G-MIDF matrix. For this reason, the G-MIDF exhibited small clumps and irregular fragments on its surface. The enzymatic extraction treatment caused structural loosening, which consequently led to increasing tendency in specific areas of the G-MIDF and hence resulted in improvements in the water holding, oil holding, and salt ion absorption capacities. 

#### 3.1.2. FTIR Analysis

The FT-IR spectra for G-IDF and G-MIDF are shown in Figure 2. The broad FTIR absorption peaks at the IR region of 3200 cm^−1^ in both the G-IDF and G-MIDF represented –OH stretching, mainly owing to the presence of cellulose, hemicellulose, and lignin. However, the FTIR peak at the IR region of 3400 cm^−1^ exhibited a diminution in the G-IDF spectrum in comparison with that of a similar peak in the case of the G-MIDF at 3400 cm^−1^ IR region. The broadness in the IR region’s intensity for the G-MIDF could be ascribed to the breakage of intermolecular linkages. The FTIR spectra peaks showed resemblance in terms of peak shapes and IR regions, which implied that enzymatic modification did not exert any significant effect on the functional groups’ structural configurations in the case of the G-MIDF as compared to that of the G-IDF. This implied that typical functional groups of insoluble cellulose and soluble pectin exhibited stability in the G-MIDF after exposure to enzymatic hydrolysis. The strong absorption peak with a high degree of broadness at the IR region of 3419 cm^−1^ was mainly attributed to the presence of native cellulose and was ascribed to the functional group stretching of O–H, characterized as the molecular bonding of uronic acid. Ginseng residue has been reported to comprise rich heteropolysaccharides (uronic acid, 4.42% content) and to be acidic in nature, as well as being rich in amino acids, protein, and mineral elements [22]. However, notable absorption was observed at the IR region of 2629 cm^−1^, which was indicative of the C–H stretching of methylene (–CH_2_) and methyl (–CH_3_) functional groups, representing the typical structural configuration of cellulosic polysaccharides compounds. A minor IR peak was observed at the IR spectral region of 1741 cm^−1^, which was mainly due to the presence of the ester carbonyl group stretching (C=O) or acetyl (−COCH_3_) functional group stretching of hemicellulose. The presence of an IR peak was also recorded at the IR region of 1626 cm^−1^, which was ascribed to the probable presence of ionized and esterified groups of galacturonic acid. A weak peak was also observed at the IR spectral region of 1516 cm^−1^, along with the adjacent presence of a minor peak at 1427 cm^−1^; these were probably indicative of the aromatic or aliphatic C–H group IR vibrational patterns of lignin. Weak IR vibrations were recorded at 1319 cm^−1^ and 1242 cm^−1^, which were mainly attributed to the typical cellulose structure and the C–O and O–H group vibration of hemicellulose. Small peaks at IR regions of 900–1200 cm^−1^ as well as 1056 cm^−1^ were ascribed to the stretching vibration of the acyl-oxygen (CO–OR) of hemicellulose and the carbonyl (C–O) stretching vibration of the guaiacyl unit of lignin. It was evident from the above results that G-MIDF exhibited typical functional group configurations for cellulose polysaccharides, including pectin, hemicellulose, cellulose, and lignin, which was consistent with the composition of ginseng DF reported in the published literature [26]. The results of the current research were in agreement with the findings reported by Hua et al. [22], who elucidated the nutritional value and extracted the ginseng insoluble dietary fiber using the AOAC method and reported the stability of the functional groups’ configuration in ginseng IDF. 

#### 3.1.3. XRD Analysis 

X-ray diffractometry was employed to determine the crystal structure of the G-IDF and G-MIDF, and the results are depicted in Figure 3. The XRD analysis showed that the G-IDF and G-MIDF exhibited two diverse and obvious diffraction peaks at the 2θ angles of 15.02° and 21.08°, which was indicative of the presence of crystalline regions of cellulosic molecules. The peak morphology of the G-IDF was significantly different as compared to that of the G-MIDF. In comparison with the G-IDF, the peak morphology of the G-MIDF exhibited split peaks at a 2θ angle of 23.71° and the emergence of sharp peaks at the 2θ angles of 28.02°, 31.78°, and 35.07°. This implied that enzymatic treatment in the case of the G-MIDF caused partial destruction of the amorphous regions of cellulose. It has also been reported in previous studies that amorphous regions in crystals might be due to the presence of cellulose, hemicellulose, and lignin [27]. Irregular weak peaks emerging at 2θ of 36.06°, 38.08°, 39.01°, and 47.07° might be attributable to the denaturation of cellulose and hemicellulose during ginseng IDF extraction by means of enzymatic hydrolysis. Similar results have been reported by Jiang et al. [3], whereby the authors investigated the effects of alkaline hydrogen peroxide (AHP) treatment on functional and structural properties of ginseng IDF, and XRD analysis showed the presence of cellulose type 1 with a double-helix structure. AHP treatment resulted in an increase in cellulose content and decreases in lignin and hemicellulose. 

### 3.2. Thermal Properties of G-IDF and G-MIDF 

Thermal properties in terms of weight loss for the G-IDF and G-MIDF were determined, and the results are depicted in Figure 4. Thermogravimetric analysis (TGA) was performed to determine the thermal behaviors of the G-IDF and G-MIDF in order to evaluate the thermal effects caused by processing, as well as the thermal stability. The G-MIDF exhibited a TG curve profile of three weight loss phases at ranges of 50–200 °C, 200–400 °C, and 400–600 °C, with varying rates of degradation (shown in the graph in brown color); this profile resembled that of the thermal behavior of rice bran IDF [28] and ginseng IDF [22]. At the start of the TGA curve in the first pyrolytic range (50–200 °C), the G-IDF and G-MIDF exhibited evaporation at 120 °C and consequent weight loss of less than 2%, which might be implied in terms of the evaporation of absorbed water from the outer sample’s surface. However, the G-IDF exhibited comparatively more weight loss compared to that of the G-MIDF at the 50–200 °C range. In the second pyrolytic range of 200–400 °C, both the G-MIDF and G-IDF exhibited varying weight loss rates, and weight loss was markedly higher in the case of the G-MIDF as compared to the G-IDF. The increasing tendency of weight loss in the case of the G-MIDF as compared to the G-IDF might be attributable to the pyrolytic degradation of soluble hemicellulosic polysaccharides and pectin. Moreover, ginseng DF components with less solubility, such as pectin and protein, contributed to the higher pyrolytic decomposition temperature [29]. In the third pyrolytic temperature range of 400–600 °C, a slower and gradual residual weight loss of 5.43% was observed, and this was ascribed to the phenomenon of the decomposition of lignin and cellulose in the ginseng IDF. It could be implied from the above results that ginseng IDF is appropriate for food processing applications involving thermal treatment at temperatures lower than 300 °C.

Plant cell walls comprise lignin networks composed of ether and ester linkages with hemicellulose, whereas cellulose is usually found in an embedded form and plays a role in the formation of fiber–polymer complexes. Lignin’s amorphous structure exhibits high pyrolysis resistance to thermal processes, resulting in the consequently reduced thermal accessibility of cellulose. Generally, non-starch polysaccharides exhibit thermal stability in the following order: lignin > cellulose > hemicellulose. Regarding the pyrolysis mechanism of cellulose, the most commonly reported pathway is reported in the literature as follows: Activated cellulose is produced from cellulose with a lower degree of polymerization through complex reactions, such as dehydration, condensation, and polymerization [30]. Then, the activated cellulose is subjected to further degradation into intermediate products via aromatization and dehydration, resulting in the production of gaseous molecules and char or the dehydration of side functional groups, whereas condensation results in the formation of levo-glucosan. All these by-products lead to the formation of the decomposition products of char and gases [31]. The reaction conditions, including pH and enzymatic treatment, for the extraction of the ginseng DF may cause a reduced protection of lignin in the cellulosic and hemicellulosic mixture, the aggravation of hemicellulose decomposition, and a reduction in the thermal stability of hemicellulose and cellulose, consequently increasing the susceptibility of hemicellulose and cellulose to pyrolysis [29]. 

### 3.3. Viscosity of G-IDF and G-MIDF

The rheological behaviors of the G-MIDF and G-IDF were evaluated, and the results are tabulated in Table 1 in terms of the apparent viscosity, the consistency coefficient, and the flow behavior index. Rheograms are also depicted in Figure 5, showing viscosity curves as a function of the shear rate. Both the G-IDF and G-MIDF exhibited apparent viscosity ($\dot{\gamma }$) values of 6.06 and 6.22 m Pa·s (Table 1), respectively. The G-IDF has relatively lower apparent viscosity in comparison with the enzymatically modified DF (G-MIDF). It is worthwhile to note that apparent viscosity is usually influenced by hydrogen bonding and charge-transfer complexes formed between polymer chains. The G-IDF and G-MIDF exhibited consistency coefficients (K) of 67.59 and 59.50 m Pa·s, respectively. The flow behavior index (n) is usually a parameter that provides information about the degree of non-Newtonian characteristics. If the flow behavior index is greater than 1, then it shows the dilatancy nature of the extracted DFs. 

Dilatant fluids are also reported in the literature as shear-thickening fluids, which usually show an increasing tendency of apparent viscosity with the corresponding rise in the shear rate [32]. The G-IDF exhibited a lower flow behavior index (lower dilatancy) of 0.40 as compared to that of the G-MIDF (0.43). As the rheograms in Figure 5 show, the G-IDF and G-MIDF were also analyzed to determine changes in viscosity as a function of the applied shear rate. Overall, it was evident from the rheograms that both the G-IDF and G-MIDF exhibited a resemblance with respect to viscosity changes as a function of shear rate, and enzymatic hydrolysis did not exert any significant (*p* > 0.05) effect on the apparent viscosity of the G-MIDF in comparison with the G-IDF. The consistency coefficient (K) value of the G-MIDF was significantly (*p* < 0.05) lower (59.51) than that of the G-IDF (67.59). This could be attributed to the destruction of ginseng DF’s molecular structure, breakage of partial branching patterns, as well as the decreasing trend in internal molecular linkages after exposure to enzymatic hydrolysis. Moreover, slight differences in the flow behavior index and apparent viscosity were observed between the G-MIDF and G-IDF. Similar findings have been reported by Jiang et al. [25] for dietary fiber extracted from Sanchi flower (*Panax notoginseng*). 

### 3.4. Hydration and Functional Properties of G-IDF and G-MIDF

The hydration and functional properties of the G-IDF and G-MIDF were evaluated, and the results are tabulated in Table 2. The following hydration and functional properties were analyzed: water holding capacity (WHC), oil retention capacity (ORC), water swelling capacity (WSC), nitrite ion binding capacity (NAC), bile acid binding capacity (BAC), cholesterol absorption capacity (CAC), and glucose absorption capacity (GAC). 

#### 3.4.1. WHC 

Enzymatic hydrolysis resulted in a significantly higher (*p* < 0.05) WHC (6.36 g/g) in case of the G-MIDF as compared to that of the WHC of the G-IDF (5.31 g/g) (Table 2). It was evident from the results that enzymatic hydrolysis caused a 20% increase in the WHC in the G-MIDF as compared to the G-IDF. The WHC has been reported to be influenced by several factors, including surface area and particle size. Similar results of rising tendencies in the WHC have been reported in the case of dietary fiber extracted from Sanchi flower (*Panax notoginseng*) [8], arrowhead tuber starch (*Sagittaria sagittifolia* L.) [33], and rice bran dietary fiber [7]. The rising tendency in the WHC can be ascribed to the probable breakdown of hydrogen bonds in the G-MIDF after enzymatic hydrolysis, which causes increased exposure of hydrophilic groups. However, from a health benefits perspective, the WHC of the ginseng DF exerts significant influence on food viscosity, which consequently affects transit time (in the small intestine), accompanied with a slowing of gastric clearance. In terms of product development, DF’s WHC may exert an indirect effect on the product’s production cost and shelf life when fiber-incorporated products are formulated, such as jams and fruit juices.

#### 3.4.2. ORC 

The ORC values of the G-MIDF and G-IDF are given in Table 2. The G-MIDF, after exposure to enzymatic hydrolysis, exhibited a rise in ORC by 55.67% from 2.91 (G-IDF) to 4.52 g/g (G-MIDF). The ORC is indicative of the fiber’s adsorption capacity for lipophilic constituents. Excluding IDF’s hydrophobicity, IDF’s surface characteristics and total charge density have been reported to exert significant influence on the ORC. An enhanced ORC is an important property with beneficial effects in meat product processing. The enzymatic hydrolysis had a significant influence on physical properties, such as the appearance of a porous and wrinkled surface, which might be helpful in oil retention. Similar results have been reported by Jiang et al. [3] in the case of ginseng IDF modified by exposure to alkaline hydrogen peroxide treatment.

#### 3.4.3. WSC 

The WSC values of the G-MIDF and G-IDF are given in Table 2. The WSC exhibited a significant (*p* < 0.05) rise by 28.85% from 6.17 g/g (G-IDF) to 7.91 g/g (G-MIDF). The increasing tendency of the WSC might be ascribed to the association of the WSC with the corresponding cellulose degradation. Enzymatic hydrolysis caused the disintegration of long cellulose chains to form small cellulose polymer chains, which causes the incorporation of enlarged void spaces. Exposure to enzymatic treatment has been reported to not only cause disruption in the covalent linkages between cellulose and hemicellulose but also to lead to the breakage of ether linkages linking hemicellulose to lignin. The results of the current research were in line with findings reported by Jiang et al. [3] for ginseng IDF subjected to alkaline hydrogen peroxide treatment. 

#### 3.4.4. Nitrite Ion Adsorption Capacity (NAC)

The NAC values of the G-MIDF and G-IDF are given in Table 3. At pH 2, the NAC exhibited a significant (*p* < 0.05) increase by 4.23% from 119.92 µg/g (G-IDF) to 124.28 µg/g (G-MIDF). Likewise, at pH 7.0, the NAC increased from 117.17 µg/g (G-IDF) to 122.29 µg/g (G-MIDF). Hence, enzymatic hydrolysis improved the NAC at both pH levels. Similarly, the increasing tendencies in NAC have also been reported by Luo et al. [20] in the case of shell fibers extracted from bamboo shoot [20]. Moreover, the authors reported the NAC of IDF to be significantly higher compared to of the NACs of SDF and TDF. DFs exhibit important characteristics of fiber’s capacity to absorb nitrite. Moreover, the utilization of nitrite ions (NO_2_^–^)has been reported in the processing of cured meat products for rendering development of unique colors and characteristic flavors, as well as pathogen inhibition [9,26]. Similar findings have been reported in the case of the adsorption capacities of papaya-seed-supplemented bread and the adsorption capacity of IDF extracted from ginseng residue. However, these nitrite ions have also been reported to play a role in the formation of *N*-nitroso compounds after coming into contact with amides and secondary amines (under acidic conditions), which exhibit carcinogenicity. The NAC is regarded as one of DF’s most essential properties, which may play a contributing role in lowering toxicity to humans. Therefore, in order to overcome this adverse effect pertaining to cured meat products, phyto-chemicals with high NACs could be employed, and G-MIDF with improved NAC might be incorporated into reformulated meat products [26]. 

#### 3.4.5. Bile Acid Binding Capacity (BAC) 

The BAC values of the G-MIDF and G-IDF are given in Table 3. Previous published reports have reported on the fact that DF’s capacity to bind bile acid has implications on the incidence cardiovascular diseases (CVD). The G-MIDF exhibited an improved BAC value, with a corresponding rise by 4.14% from 90.60 mg/g (G-IDF) to 93.34 mg/g. DF’s increased binding to BAs results in the augmentation of BA’s elimination in conjunction with cholesterol–bile acid conversion, hence leading to lowering of the serum cholesterol level. These results were in agreement with the findings reported by previous studies, whereby SDF extracted by means of acidic treatment and homogenization from pineapple pomace exhibited a higher BAC as compared to that of SDFs extracted by enzymatic and ultrasonication treatments [1]. The increased BAC of the G-MIDF could be ascribed to the probable changes in FTIR absorption peaks, which were indicative of alterations in surface characteristics. Furthermore, the anionic group content and DF’s gel characteristics have also been reported to exert significant influence on the BAC. Conclusively, SDF’s internal structural and surface properties as well as particle sizes also have contributory roles affecting the BAC.

#### 3.4.6. Cholesterol Absorption Capacity (CAC)

The CAC is regarded as an important parameter for the evaluation of IDF’s ability to absorb lipophilic components both under conditions of the intestinal environment and in stimulated stomachs. The CAC values of the G-MIDF and G-IDF are provided in Table 3. The CAC exhibited a significant (*p* < 0.05) rise by 24.15% from 8.86 mg/g (G-IDF) to 10.85 mg/g (G-MIDF). After exposure to the enzymatic hydrolysis, the G-MIDF exhibited significant improvement in the CAC. The possible reason for the enhanced CAC in case of the G-MIDF might be ascribed to the loosening of the IDF’s structural framework due to enzymatic hydrolysis, which led to an increased degree of exposure of hydrophobic groups in comparison with the G-IDF. Furthermore, it may be implied that the higher CAC values of the G-MIDF were because of rising tendencies in various properties of the G-MIDF, including specific surface area and viscosity, after exposure to enzymatic hydrolysis, which led to the formation of more complex structures and resulted in the increased binding of cholesterol [34,35]. These results were in line with the findings reported for bamboo shoot fiber [20], peach fiber [36], and wheat bran [37]. 

#### 3.4.7. Glucose Absorption Capacity (GAC) 

The GAC values of the G-MIDF and G-IDF are given in Table 3. At 50 mmol/L, the GAC values for the G-IDF and G-MIDF were evaluated, and the G-MIDF showed rises in GAC values by 5.16% from 16.49 mg/g (G-IDF) to 17.34 mg/g. Higher GAC values were indicative of the increasing tendency of the IDF with respect to transit time in the gastrointestinal tract. At 100 mmol/L, a rise in GAC values by 10.45% from 31.97 mg/g (G-IDF) to 35.31 mg/g (G-MIDF) was observed. Moreover, at 150 mmol/L, the G-MIDF exhibited a rise in GAC value by 4.91% from 52.62 mg/g as compared to that of 50.16 mg/g in the G-IDF. Similarly, in a study reported on the IDF of buckwheat, increased GAC values were reported to be correlated with the corresponding increased glucose concentration. In this research, the results showed that G-MIDF exhibited a significantly high (*p* < 0.05) absorption ability in terms of binding cholesterol after exposure to enzymatic hydrolysis at all concentrations. The probable mechanism for enhanced GAC might be ascribed to glucose absorption by IDF, which may possibly occur because of the physical barrier as well as the chemisorption and ensnarement of glucose molecules. Most probably, the enhanced availability of cavities and surface areas due to the structural changes caused by enzymatic hydrolysis may help in achieving a greater entrapment of glucose molecules [25]. Furthermore, increased entrapment is also aided by the increased availability of polar groups resulting from the enzymatic treatment of DFs. Conclusively, it may be implied that an improved GAC makes ginseng DF ideal and appropriate for formulating dietary snacks as a low-calorie ingredient. Similar results have been reported by Jiang et al. [3] in the case of ginseng IDF modified by exposure to alkaline hydrogen peroxide treatment.

## 4. Conclusions

This study investigated the effects of enzymatic modification using cellulase/xylanase on the composition and structural and functional properties of ginseng insoluble dietary fiber (IDF). Fourier transform infrared spectroscopy, X-ray diffractometry, and scanning electron microcopy showed significant structural alterations in the ginseng-modified (G-MIDF) samples, which exhibited more porous and completely wrinkled surfaces. Thermo-gravimetric analysis showed that the G-MIDF exhibited a specified range of pyrolysis temperatures, making it suitable for food applications involving processing at temperatures below 300 °C. The G-IDF had relatively lower apparent viscosity in comparison with the enzymatically modified DF (G-MIDF). Overall, it was evident from the rheograms that both the G-IDF and G-MIDF exhibited a resemblance with respect to viscosity changes as a function of shear rate. Enzymatic treatment led to significant improvement in hydration and functional properties, such as water holding, oil retention, water swelling, nitrite ion binding, bile acid binding, cholesterol absorption, and glucose absorption capacities.

## Figures and Tables

**Figure 1 foods-12-02809-f001:**
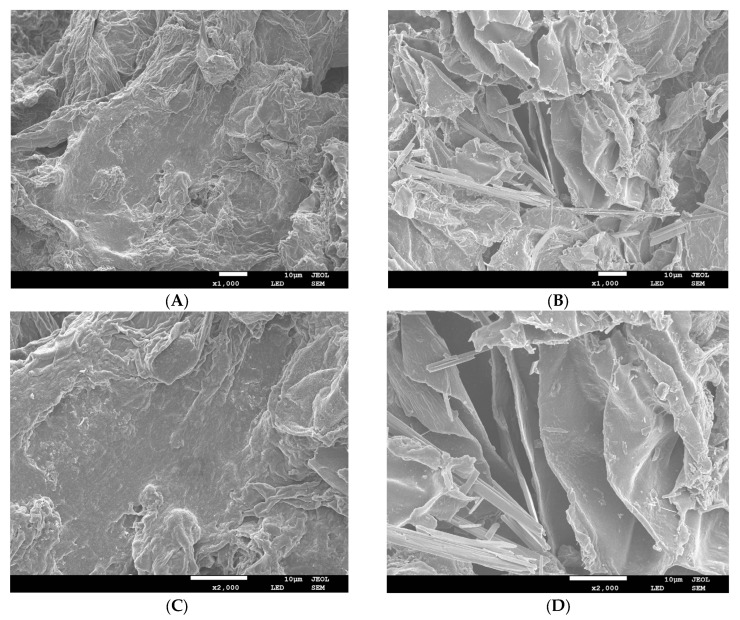
SEM images of G-IDF ((**A**): ×1000 magnification; (**C**): ×2000 magnification) and G-MIDF ((**B**): ×1000 magnification; (**D**): ×2000 magnification). Ginseng insoluble dietary fiber (G-IDF); ginseng-modified insoluble dietary fiber (G-MIDF).

**Figure 2 foods-12-02809-f002:**
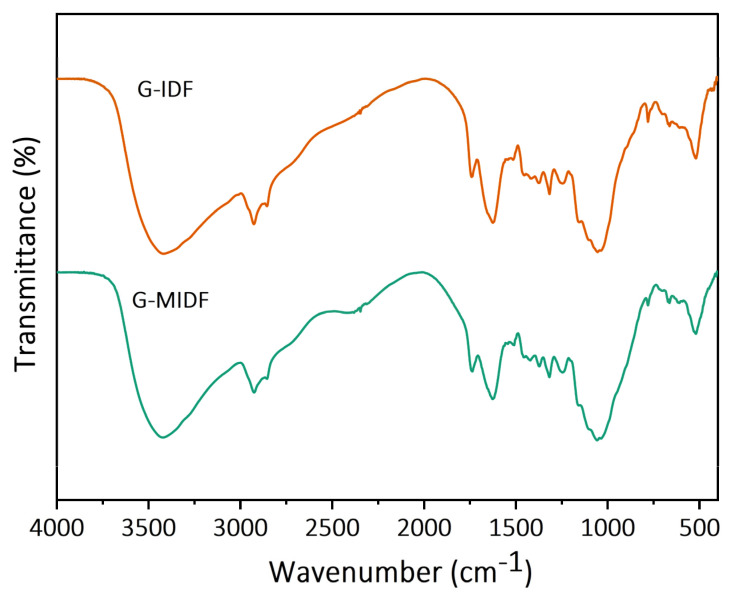
FT-IR spectra for G-IDF and G-MIDF. Ginseng insoluble dietary fiber (G-IDF); ginseng-modified insoluble dietary fiber (G-MIDF).

**Figure 3 foods-12-02809-f003:**
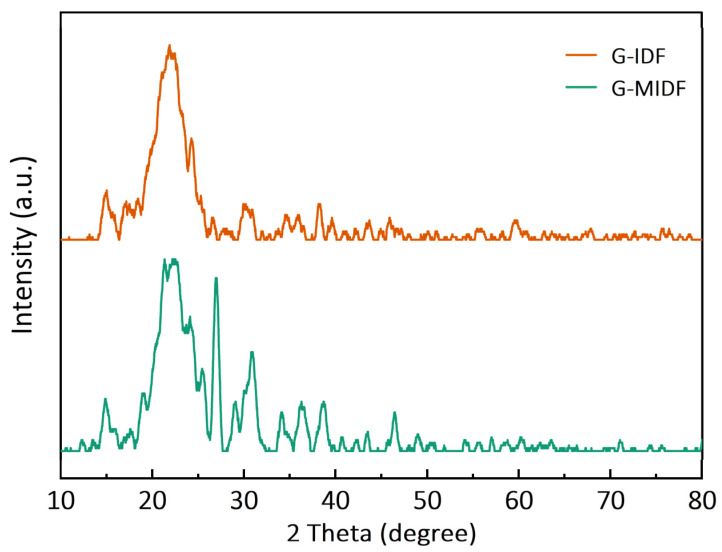
XRD pattern of G-IDF and G-MIDF. Ginseng insoluble dietary fiber (G-IDF); ginseng-modified insoluble dietary fiber (G-MIDF).

**Figure 4 foods-12-02809-f004:**
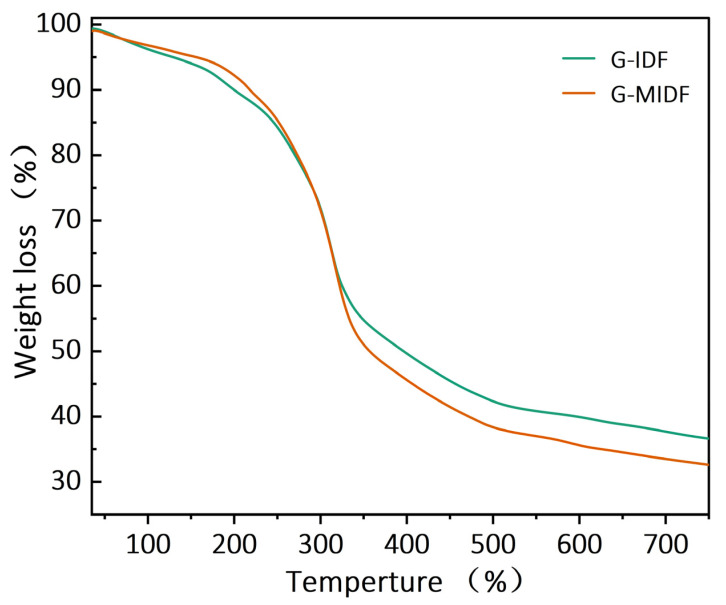
Thermal properties for G-IDF and G-MIDF. Ginseng insoluble dietary fiber (G-IDF); ginseng-modified insoluble dietary fiber (G-MIDF).

**Figure 5 foods-12-02809-f005:**
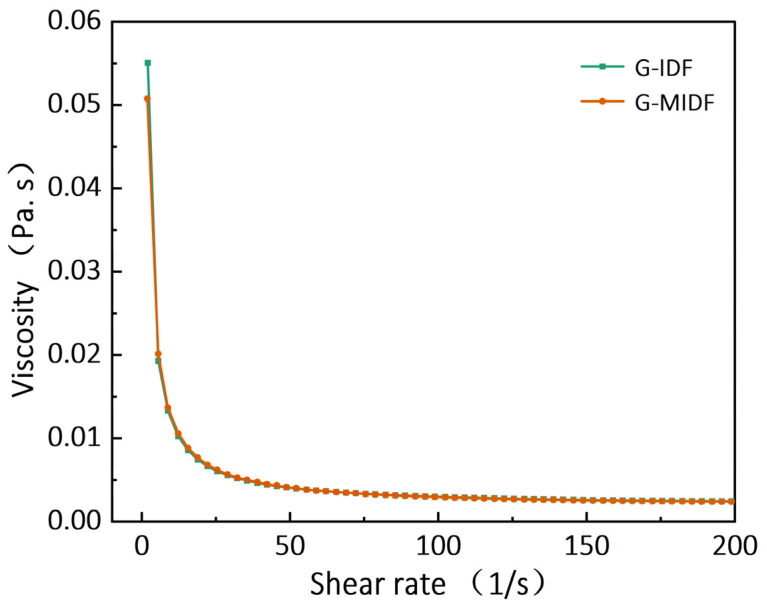
Rheogram plots of G-IDF and G-MIDF depicting the shear rate. Ginseng insoluble dietary fiber (G-IDF).

**Table 1 foods-12-02809-t001:** Consistency coefficient, flow behavior index, and apparent viscosity of G-IDF and G-MIDF.

Parameter	G-IDF	G-MIDF
Apparent viscosity 25 1/s ($\dot{\gamma }$, mPa·s)	6.06 ± 0.10 a	6.22 ± 0.04 a
Consistency coefficient (K, mPa·s)	67 ± 1.24 a	59 ± 1.75 b
Flow behaviour index (n, -)	0.40 ± 0.002 b	0.43 ± 0.012 a

Means ± SD (n = 3) with different letters (a, b) indicating significant difference (*p* < 0.05). Ginseng insoluble dietary fiber (G-IDF); Ginseng-modified insoluble dietary fiber (G-MIDF).

**Table 2 foods-12-02809-t002:** Hydration properties of G-IDF and G-MIDF.

Property	G-IDF	G-MIDF
WHC (g/g)	5.30 ± 0.09 b	6.36 ± 0.10 a
ORC (g/g)	2.91 ± 0.25 b	4.52 ± 0.06 a
WSC (g/g)	6.17 ± 0.06 b	7.90 ± 0.17 a

Means ± SD (n = 3) with different letters (a, b) indicating significant difference (*p* < 0.05). Ginseng insoluble dietary fiber (G-IDF); ginseng-modified (G-MIDF) insoluble dietary fiber. WHC: Water holding capacity; ORC: oil retention capacity; WSC: water swelling capacity.

**Table 3 foods-12-02809-t003:** Functional properties of G-IDF and G-MIDF.

Fiber Type	NAC (μg/g)		BAC (mg/g)	CAC (mg/g)	GAC (mg/g)		
	pH 2.0	pH 7.0			50 mmol/L	100 mmol/L	150 mmol/L
G-IDF	119.92 ± 0.22 b	117.17 ± 0.38 b	90.60 ± 0.23 b	8.86 ± 0.04 b	16.49 ± 0.25 b	31.97 ± 0.07 b	50.16 ± 0.04 b
G-MIDF	124.28 ± 0.15 a	122.29 ± 0.30 a	93.34 ± 0.33 a	10.85 ± 0.07 a	17.34 ± 0.19 a	35.31 ± 0.12 a	52.62 ± 0.06 a

Means ± SD (n = 3) with different letters (a, b) indicating significant difference (*p* < 0.05). Ginseng insoluble dietary fiber (G-IDF); ginseng-modified insoluble dietary fiber (G-MIDF).

## Data Availability

Data is contained within the article.

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
