# Peer review of "Modification of Ginseng Insoluble Dietary Fiber by Enzymatic Method: Structural, Rheological, Thermal and Functional Properties"

_foods, 2023, doi:10.3390/foods12142809_

Round 1

Reviewer 1 Report

The work assessed the impact of enzymatic modification on ginseng dietary fiber. The process altered the fiber's structure, making it porous and wrinkled, and enhanced its thermo-stability, making it suitable for food processing below 300ºC. Additionally, the modification improved the fiber's ability to hold water, retain oil, swell, bind to nitrite ions and bile acid, and absorb cholesterol and glucose. My major concern regarding the work is the lack of variety regarding the experimental conditions. The authors only produced one sample. If this is ok, the work should correct small typos. 

Line 96: please insert the number of hours

Line 151: Please put the -1 in superscript

Line 101: Please make some comment or citation to support these conditions of modification.  

Please correct Table 1

Where

Consistency coefficient [K, mPa s] 67.59±1.24a / 59.50±1.75b

to

Consistency coefficient [K, mPa s] 67 ± 1.24a / 59 ± 1.75 b

Author Response

Reviewer 1

The work assessed the impact of enzymatic modification on ginseng dietary fiber. The process altered the fiber's structure, making it porous and wrinkled, and enhanced its thermo-stability, making it suitable for food processing below 300ºC. Additionally, the modification improved the fiber's ability to hold water, retain oil, swell, bind to nitrite ions and bile acid, and absorb cholesterol and glucose. My major concern regarding the work is the lack of variety regarding the experimental conditions. The authors only produced one sample. If this is ok, the work should correct small typos.

Response: Ginseng is the representative sample of Panax genus. Sample representativeness is also obvious from these of our published reports. We have already published on Ginseng other varieties. Reviewer may have look over these as mentioned below: In order to make the sample representative for this study, other varieties have not been taken into account in this study. Moreover, the fiber content and the consumer acceptance of Ginseng specie and variety reported in this study is the highest as compared to that of other varieties. The methods reported in current study are not used for the first time in current research work. These methods have already been optimized by other researchers in previously published reports as given below: However, to facilitate the readers and reviewers, more pertinent details have been added in revised version of manuscript. 

  • Jiang, G., Wu, Z., Ameer, K., Li, S., & Ramachandraiah, K. (2020). Particle size of ginseng (Panax ginseng Meyer) insoluble dietary fiber and its effect on physicochemical properties and antioxidant activities. Applied Biological Chemistry63, 1-10.
  • Jiang, G., Ramachandraiah, K., Wu, Z., & Ameer, K. (2022). The Influence of Different Extraction Methods on the Structure, Rheological, Thermal and Functional Properties of Soluble Dietary Fiber from Sanchi (Panax notoginseng) Flower. Foods11(14), 1995.
  • Jiang, G., Feng, X., Wu, Z., Li, S., Bai, X., Zhao, C., & Ameer, K. (2021). Development of wheat bread added with insoluble dietary fiber from ginseng residue and effects on physiochemical properties, in vitro adsorption capacities and starch digestibility. LWT149, 111855.
  • Huo, N., Ameer, K., Wu, Z., Yan, S., Jiang, G., & Ramachandraiah, K. (2022). Preparation, characterization, structural analysis and antioxidant activities of phosphorylated polysaccharide from Sanchi (Panax notoginseng) flower. Journal of Food Science and Technology59(12), 4603-4614.
  • Wu, Z., Ameer, K., & Jiang, G. (2021). Effects of superfine grinding on the physicochemical properties and antioxidant activities of Sanchi (Panax notoginseng) flower powders. Journal of Food Science and Technology58(1), 62-73.
  • Jiang, G., Ramachandraiah, K., Murtaza, M. A., Wang, L., Li, S., & Ameer, K. (2021). Synergistic effects of black ginseng and aged garlic extracts for the amelioration of nonalcoholic fatty liver disease (NAFLD) in mice. Food Science & Nutrition9(6), 3091-3099.

Line 96: please insert the number of hours

Response: Added

Line 151: Please put the -1 in superscript

Response: Corrected

Line 101: Please make some comment or citation to support these conditions of modification.  

Response: Corrected

Please correct Table 1

Where

Consistency coefficient [K, mPa s] 67.59±1.24a / 59.50±1.75b

to

Consistency coefficient [K, mPa s] 67 ± 1.24a / 59 ± 1.75 b

Response: Corrected

Reviewer 2 Report

Thank you for submitting the manuscript "Modification of Ginseng Insoluble Dietary Fiber by Enzymatic Method: Structural, Rheological, Thermal and Functional Properties" to Foods. In general, the research is interesting and up-to-date, although it includes only one method of modifying insoluble fiber (enzymatic) and therefore it is not possible to know whether this method is really the most suitable for this material. In addition, the authors did not test many enzymatic treatments and used only one published method, which leaves another limitation for the work in relation to the number of treatments tested. 

Although the manuscript brings a consistent physical and technological characterization of the fibrous hydrolysate, the discussions are somewhat difficult due to the lack of chemical characterization (determination of carbohydrates, for example).

Line#14: I've never seen this happen because these analyzes are not quantitative, how was their statistical analysis done?

Introduction: it is necessary to make clear the importance of using this by-product. What is the annual generation of ginseng waste? What is its economic importance?

Line#234: You cannot make this claim if the pectin, cellulose or hemicellulose content has not been studied.

Line#237 and #238: These statements need to be referenced.

All captions for figures and tables must define the abbreviations, as the reading of these materials is independent of the reading of the text.

English Language is fine.

Author Response

Reviewer 2

Thank you for submitting the manuscript "Modification of Ginseng Insoluble Dietary Fiber by Enzymatic Method: Structural, Rheological, Thermal and Functional Properties" to Foods. In general, the research is interesting and up-to-date, although it includes only one method of modifying insoluble fiber (enzymatic) and therefore it is not possible to know whether this method is really the most suitable for this material. In addition, the authors did not test many enzymatic treatments and used only one published method, which leaves another limitation for the work in relation to the number of treatments tested. 

Response: Ginseng is the representative sample of Panax genus. Sample representativeness is also obvious from these of our published reports. We have already published on Ginseng other varieties. Reviewer may have look over these as mentioned below: In order to make the sample representative for this study, other varieties have not been taken into account in this study. Moreover, the fiber content and the consumer acceptance of Ginseng specie and variety reported in this study is the highest as compared to that of other varieties. The other extraction and modification methods have been already exploited for ginseng insoluble dietary fiber. The methods reported in current study are not used for the first time in current research work. These methods have already been optimized by other researchers in previously published reports as given below: However, to facilitate the readers and reviewers, more pertinent details have been added in revised version of manuscript. 

  • Jiang, G., Wu, Z., Ameer, K., Li, S., & Ramachandraiah, K. (2020). Particle size of ginseng (Panax ginseng Meyer) insoluble dietary fiber and its effect on physicochemical properties and antioxidant activities. Applied Biological Chemistry63, 1-10.
  • Jiang, G., Ramachandraiah, K., Wu, Z., & Ameer, K. (2022). The Influence of Different Extraction Methods on the Structure, Rheological, Thermal and Functional Properties of Soluble Dietary Fiber from Sanchi (Panax notoginseng) Flower. Foods11(14), 1995.
  • Jiang, G., Feng, X., Wu, Z., Li, S., Bai, X., Zhao, C., & Ameer, K. (2021). Development of wheat bread added with insoluble dietary fiber from ginseng residue and effects on physiochemical properties, in vitro adsorption capacities and starch digestibility. LWT149, 111855.
  • Huo, N., Ameer, K., Wu, Z., Yan, S., Jiang, G., & Ramachandraiah, K. (2022). Preparation, characterization, structural analysis and antioxidant activities of phosphorylated polysaccharide from Sanchi (Panax notoginseng) flower. Journal of Food Science and Technology59(12), 4603-4614.
  • Wu, Z., Ameer, K., & Jiang, G. (2021). Effects of superfine grinding on the physicochemical properties and antioxidant activities of Sanchi (Panax notoginseng) flower powders. Journal of Food Science and Technology58(1), 62-73.
  • Jiang, G., Ramachandraiah, K., Murtaza, M. A., Wang, L., Li, S., & Ameer, K. (2021). Synergistic effects of black ginseng and aged garlic extracts for the amelioration of nonalcoholic fatty liver disease (NAFLD) in mice. Food Science & Nutrition9(6), 3091-3099.

Although the manuscript brings a consistent physical and technological characterization of the fibrous hydrolysate, the discussions are somewhat difficult due to the lack of chemical characterization (determination of carbohydrates, for example).

Response: The polysaccharides content of ginseng is reported to be 43%.The ginseng polysaccharides content has already reported in published literature. In this study, proximate composition was not calculated, however, keeping in view the reviewer’s appreciable suggestion, we shall incorporate the proximate composition.

Zhang, J. K., Gao, R., Dou, D. Q., & Kang, T. G. (2013). The ginsenosides and carbohydrate profiles of ginseng cultivated under mountainous forest. Pharmacognosy Magazine9(Suppl 1), S38.

In this study, FTIR was carried out and functional groups were elucidated. The FTIR spectra peaks showed resemblance in terms of peak shapes and IR regions, which implied that enzymatic modification did no exert any significant effect on the functional groups structural configuration in case of G-MIDF as compared to that of G-IDF. This implied that typical functional groups of insoluble cellulose and soluble pectin exhibited stability in G-MIDF after exposure to enzymatic hydrolysis. The strong absorption peak with high degree of broadness at IR region of 3419 cm-1 was mainly attributed to the presence of native cellulose and was ascribed to the functional group stretching of O–H characterized as molecular bonding of uronic acid. Ginseng residue has been reported to comprise of rich heteropolysaccharide (uronic acid, 4.42% content) acidic in nature being rich in amino acids, protein and mineral elements.

Line#14: I've never seen this happen because these analyzes are not quantitative, how was their statistical analysis done?

Response: This was typographical mistake and it was not statistical but it was actually obvious. Revised as per reviewer’s suggestion. We are thankful to reviewer.

Introduction: it is necessary to make clear the importance of using this by-product. What is the annual generation of ginseng waste? What is its economic importance?

Response: The annual generation of ginseng waste is difficult to estimate, as there is no centralized data collection system. However, some estimates suggest that the global annual generation of ginseng waste is in the range of 100,000 to 200,000 tons. This waste is generated from a variety of sources, including the cultivation, processing, and transportation of ginseng. The composition of ginseng waste varies depending on the source. However, it typically includes plant material, such as roots, leaves, and stems, as well as processing byproducts, such as sawdust and water.

  • Temizel, Ä°., Emadian, S. M., Di Addario, M., Onay, T. T., Demirel, B., Copty, N. K., & Karanfil, T. (2017). Effect of nano-ZnO on biogas generation from simulated landfills. Waste management63, 18-26.
  • Ma, X., Armas, S. M., Soliman, M., Lytle, D. A., Chumbimuni-Torres, K., Tetard, L., & Lee, W. H. (2018). In situ monitoring of Pb2+ leaching from the galvanic joint surface in a prepared chlorinated drinking water. Environmental science & technology52(4), 2126-2133.
  • Chen, W., Balan, P., & Popovich, D. G. (2020). Changes of ginsenoside composition in the creation of black ginseng leaf. Molecules25(12), 2809.
  • Yang, Y., Ren, C., Zhang, Y., & Wu, X. (2017). Ginseng: an nonnegligible natural remedy for healthy aging. Aging and disease8(6), 708.

It is a non-starch polysaccharide that is made up of cellulose, hemicellulose, and lignin. Ginseng IDF has a number of health benefits. Ginseng IDF can help to regulate bowel movements and promote the growth of beneficial bacteria in the gut. This can help to improve digestion and reduce the risk of constipation, diarrhea, and other digestive problems [1]. Ginseng IDF can help to lower LDL (bad) cholesterol levels and raise HDL (good) cholesterol levels. This can help to reduce the risk of heart disease, stroke, and other cardiovascular problems. Ginseng IDF can help to slow down the absorption of glucose into the bloodstream. This can help to keep blood sugar levels stable and reduce the risk of type 2 diabetes [2]. Ginseng IDF can help to boost the immune system by stimulating the production of white blood cells. This can help to protect the body against infection. Ginseng IDF can help to reduce inflammation by blocking the production of pro-inflammatory cytokines. This can help to improve symptoms of conditions such as arthritis, asthma, and allergies [3]. In addition to these health benefits, ginseng IDF is also a good source of fiber, which is essential for overall health. Fiber helps to keep the digestive system healthy, promote weight loss, and regulate blood sugar levels. Ginseng IDF is a safe and effective dietary supplement that can be added to the diet in a number of ways. It can be found in capsules, powders, and teas. It can also be added to smoothies, yogurt, and other foods [4]. Ginseng IDF may interact with certain medications, so it is important to make sure that it is safe for you to take. Overall, ginseng IDF is a beneficial dietary fiber that has a number of health benefits. It can help to improve gut health, reduce cholesterol levels, lower blood sugar levels, boost the immune system, and reduce inflammation. If you are looking for a way to improve your health, adding ginseng IDF to your diet is a good option [5].

  1. Hua, M.; Liu, Z.; Sha, J.; Li, S.; Dong, L.; Sun, Y. Effects of ginseng soluble dietary fiber on serum antioxidant status, immune factor levels and cecal health in healthy rats. Food Chemistry 2021, 365, 130641.
  2. Hua, M.; Fan, M.; Li, Z.; Sha, J.; Li, S.; Sun, Y. Ginseng soluble dietary fiber can regulate the intestinal flora structure, promote colon health, affect appetite and glucolipid metabolism in rats. Journal of Functional Foods 2021, 83, 104534.
  3. Jackson, C.; Rupasinghe, H.V.; Ayanbadejo, T.; Martos, P.; Schooley, J. Assessment of Ontario-grown ginseng (Panax quinquefolius L.) for nutritional quality and food safety. In Proceedings of the XXVI International Horticultural Congress: The Future for Medicinal and Aromatic Plants 629, 2002; pp. 115-121.
  4. Yu, H.-Y.; Rhim, D.-B.; Kim, S.-K.; Ban, O.-H.; Oh, S.-K.; Seo, J.; Hong, S.-K. Growth promotion effect of red ginseng dietary fiber to probiotics and transcriptome analysis of Lactiplantibacillus plantarum. Journal of Ginseng Research 2023, 47, 159-165.
  5. Jeon, H.J.; You, S.-H.; Nam, E.H.; Truong, V.-L.; Bang, J.-H.; Bae, Y.-J.; Rarison, R.H.; Kim, S.-K.; Jeong, W.-S.; Jung, Y.H. Red ginseng dietary fiber promotes probiotic properties of Lactiplantibacillus plantarum and alters bacterial metabolism. Frontiers in Microbiology 2023, 14, 1139386.

Line#234: You cannot make this claim if the pectin, cellulose or hemicellulose content has not been studied.

Response: Corrected as per reviewer’s suggestion.

Line#237 and #238: These statements need to be referenced.

Response: Corrected as per reviewer’s suggestion.

All captions for figures and tables must define the abbreviations, as the reading of these materials is independent of the reading of the text.

Response: Corrected as per reviewer’s suggestion.
